# An experimental seasonal hydrological forecasting system over the Yellow River basin-Part I: Understanding the role of initial hydrological conditions

Xing Yuan[1], Feng Ma[1,2], Linying Wang[1,3], Ziyan Zheng[1], Zhuguo Ma[1], Aizhong Ye[2], and Shaoming Peng[4]

[1]RCE-TEA, Institute of Atmospheric Physics, Chinese Academy of Sciences, Beijing, 100029, China
[2]State Key Laboratory of Earth Surface Processes and Resource Ecology, College of Global Change and Earth System Science, Beijing Normal University, Beijing, 100875, China
[3]University of Chinese Academy of Sciences, Beijing, 100049, China
[4]Yellow River Engineering Consulting Co. Ltd., Zhengzhou, 450003, China

*Correspondence to*: Xing Yuan (yuanxing@tea.ac.cn)

**Abstract.** The hydrological cycle over the Yellow River has been altered by the climate change and human interventions greatly during past decades, with a decadal drying trend mixed with a large variation of seasonal hydrological extremes. To provide support for the adaptation to a changing environment, an experimental seasonal hydrological forecasting system is established over the Yellow River basin. The system draws from a legacy of a global hydrological forecasting system that is able to make use of real-time seasonal climate predictions from North American Multimodel Ensemble (NMME) climate models through a statistical downscaling approach, but with a higher resolution and a spatially disaggregated calibration procedure that are based on a newly compiled hydrological observation dataset with five decades of naturalized streamflow at twelve mainstream gauges and a newly released meteorological observation dataset including 324 meteorological stations over the Yellow River basin. While the evaluation of the NMME-based seasonal hydrological forecasting will be presented in a companion paper to explore the added values from climate forecast models, this paper investigates the role of initial hydrological conditions (ICs) by carrying out six-month Ensemble Streamflow Prediction (ESP) and reverse ESP-type simulations for each calendar month during 1982-2010 with the hydrological models in the forecasting system, i.e., a large-scale land surface hydrological model and a global routing model that is regionalized over the Yellow River. In terms of streamflow predictability, the ICs outweigh the meteorological forcings up to 2-5 months during the cold and dry seasons, but the latter prevails over the former in the predictability after the first month during the warm and wet seasons. For the streamflow forecasts initialized at the end of the rainy season, the influence of ICs for lower reaches of the Yellow River can be 5 months longer than that for the upper reaches, while such difference drops to 1 month during the rainy season. Based on an additional ESP-type simulation without the initialization of the river routing model, it is found that the initial surface water state is the main source of streamflow predictability during the first month, beyond which other sources of terrestrial memory become more important. During the dry/wet periods, the dominance of ICs on the streamflow predictability can be extended by a month even in the rainy season, suggesting the usefulness of the ESP forecasting approach after the onset of

the hydrological extreme events. Similar results are found for the soil moisture predictability, but with longer influences from ICs. And the simulations indicate that the soil moisture memory is longer over the middle reaches than those over the upper and lower reaches of the Yellow River. The naturalized hydrological predictability analysis in this study will provide a guideline for establishing an operational hydrological forecasting system as well as for managing the risks of hydrological

extremes over the Yellow River basin.

1.  **Introduction**

Global warming has fundamentally affected terrestrial hydrological cycle, as well as water-related sectors. The intensification of the water cycle leads to an increase of hydrological extreme events such as flooding and droughts, which

influences the reservoir regulation and flood mitigation, and the coordination of the water supply for agricultural, urban and environmental sustainability (Huntington, 2006; Oki and Kanae, 2006). While the mitigation activities including the reduction of carbon emission will not have a significant impact on slowing the global warming until a few decades later due to the inertia of the climate system, the adaptation can be an approach that reduces the negative effects from climate change in a timely manner (IPCC, 2014). Nevertheless, a well-planned adaptation cannot be achieved without a reliable prediction

of the future.

In terms of terrestrial hydrology, a basic question is how to manage the water resources that is adaptive to climate change, especially to the extreme events (e.g., droughts). In other words, how to predict the future hydrology at a lead time that is not only long enough for taking an action, but also reliable for an effective adaptation is a big concern both for science and application. While the decadal hydrological prediction is still at an exploring stage due to very limited predictability over

land, the seasonal hydrological forecasting has been carried out for about half a century (Pagano et al., 2004), and are being augmented with physical hydrological models (Day, 1985; Bierkens and van Beek, 2009; van Dijk et al., 2013; Svensson et al., 2015) through the Ensemble Streamflow Prediction (ESP) method as well as climate forecast models (Wood et al., 2002; Luo and Wood, 2008; Mo and Lettenmaier, 2014; Yuan et al., 2013, 2015a) where the climate predictions are downscaled to drive the physical hydrological models and provide the hydrological forecasting (Yuan et al., 2015b).

Statistical, dynamical and hybrid seasonal hydrological forecasting systems are being developed and implemented by multiple research institutions and operational centers around the world. For example, a national seasonal streamflow forecasting service operated by the Australian Bureau of Meteorology (http://www.bom.gov.au/water/ssf), with a statistical forecasting method based on the joint distribution of future streamflow and their predictors such as antecedent streamflow and El Niño Southern Oscillation (ENSO) indices, now provides forecasts over 160 locations including major water storages

and river systems across Australia (Wang et al., 2009). A Drought Monitoring and Hydrologic Forecasting system developed by the Princeton University (http://hydrology.princeton.edu/forecast), which is based on downscaled climate prediction from the Climate Forecast System version 2 (CFSv2) and a distributed hydrological model (Luo and Wood, 2008; Yuan et al., 2013), is providing soil moisture forecasts and drought outlook over conterminous United States up to six months, and is being augmented with remote sensing products and multiple climate forecast models to provide seasonal hydrological

forecasting over Africa (Sheffield et al., 2014) and global major river basins (Yuan et al., 2015a). Besides the statistical and dynamical forecasting systems, a hybrid system called Hydrological Outlook UK (http://www.hydoutuk.net) is being developed over the Great Britain by an expert-merging of a statistical analogue and persistence method, the ESP method, and a hydrological modeling system driven by the UK Met Office climate forecasts.

Similar to the seasonal climate prediction, the initial hydrological condition (IC) is also an important source of predictability for the hydrology at seasonal time scale, and should be carefully treated in developing a hydrological forecasting system. Basically, IC of snow controls the seasonal hydrological variations significantly over the headwaters region of a river basin, especially over high altitude areas. For instance, Koster et al. (2010) found that the hydrological simulations with the IC of snow can explain up to 50% of the variance for the streamflow over the western USA in the following five months. The
importance of snow for streamflow predictability was also confirmed over European basins (Staudinger and Seibert, 2014) and global major river basins (Yossef et al., 2013). As compared with snow, soil moisture has less impact on the hydrological predictability during the snow melting season, but can affect the predictability significantly during other seasons, where its dominance can last over six months over certain river basins (Mahanama et al., 2012). In addition, the IC of groundwater is also important during the low-flow period where the subsurface runoff dominates the streamflow (Paiva et
al., 2012).

To assess the contributions of ICs and meteorological forcings, a theoretical framework called reverse ESP (revESP) was proposed by Wood and Lettenmaier (2008). For the ESP method, a hydrological model with realistic ICs is forced by an ensemble of meteorological forcings resampled from the history; while for the revESP, the hydrological model is driven by observed meteorological forcings (a perfect meteorological forecast), with ICs resampled from the history. Wood and
Lettenmaier (2008) applied the assessment framework over two river basins in the western USA and found that ICs yield streamflow forecasting skill for up to 5 months over northern California during the transition period between the wet and dry seasons, but have less impact over southern Colorado basin due to a weaker annual cycle of precipitation. Since then, the revESP framework has been widely used to assess the role of ICs at regional to global scales (Li et al., 2009; Koster et al., 2010; Shukla and Lettenmaier, 2011; Paiva et al., 2012; Singla et al., 2012; Shukla et al., 2013; Yossef et al., 2013;
Staudinger and Seibert, 2014; Yang et al., 2014). However, most assessments did not explicitly investigate the role of the IC of the surface water state variables in the streamflow forecasting, where it could be a major source of hydrological forecast uncertainty over rivers with low slope and large floodplains (Paiva et al., 2012). In addition, the ICs may have different impacts on the hydrological forecasting over upper and lower reaches of a large river basin, which is also important for a coordinated water resource management across the runoff generation and consumption regimes.

As the first paper of a two-part series, this paper introduces an experimental seasonal hydrological forecasting system developed over the Yellow River basin in northern China, and investigates the hydrological predictability across the main stream of the Yellow River. The revESP method is used to assess the contributions from ICs and meteorological forcings, and the assessments conditional on the surface and subsurface water state variables, and the dry/wet conditions are being

investigated. Seasonal hydrological forecasting with multiple climate forecast models will be evaluated in a companion paper, by comparing with the ESP-based hydrological forecasting (Yuan, 2016).

## 2. Data and Method

### 2.1 Data and study domain

The Yellow River is the second longest and the second largest river in China, with a length of about 5500 km and a drainage area of $7.52\times10^5$ km$^2$. Figure 1 shows the locations of 324 meteorological stations and 12 mainstream hydrological gauges within the Yellow River, and Table 1 lists the latitude, longitude and drainage area for the 12 gauges. The Yellow River originates from the Qinghai-Tibet Plateau, wanders through the northern semiarid region including the loess plateau, passes through the eastern low land areas, and finally discharges into the Bohai Gulf (Yang et al., 2004).

The meteorological forcing datasets from 324 meteorological stations are interpolated into 1321 grids at a 0.25-degree resolution, with a lapse rate correction for temperature at different elevations. The observations from three nearest meteorological stations are interpolated to each grid by using the inverse quadratic distance weighting method. Note that the meteorological dataset compiled in this study has more regional information as compared with previous studies that are based on about 100 meteorological stations over the Yellow River basin (Yang et al., 2004; Cong et al., 2009). Figure 2

shows the gridded seasonal mean surface air temperature and precipitation averaged during 1982-2014, indicating a typical monsoon climate with hot and wet summer, and cold and dry winter. The Yellow River flows across nine provinces in China, where the upstream of the Tangnaihai gauge (Fig. 1) is the headwaters region, with a cold (Figs. 2a-d) and humid climate (Fig. 2e-h). The northwestern region between the Lanzhou and Hekouzhen gauges (Fig. 1) is a semiarid region, with low rainfall but high temperature (Fig. 2). The northeastern region between the Hekouzhen and Longmen (Fig. 1) is located in

the loess plateau, again a semiarid area. There are several main tributaries between the Longmen and Sanmenxia gauges, including the Weihe, Jinghe and Fenhe rivers (Fig. 1). The downstream of the Huayankou gauge is located in the alluvial plain, where the riverbed is elevated above the adjacent floodplains due to sediment deposition and man-made levees (Fig. 1).

### 2.2 Description of the seasonal hydrological forecasting system

Figure 3 shows the flowchart for the experimental seasonal hydrological forecasting system. The system makes use of the

seasonal climate prediction of precipitation and temperature from multiple climate forecast models participating in the North American Multimodel Ensemble (NMME) project (Kirtman et al., 2014), a spatial downscaling and bias correction method (Wood et al., 2002) that is used to transfer global climate prediction of meteorological forcings for driving a land surface hydrological model and a routing model at river basin scale. The soil moisture and streamflow predicted by the system a few months ahead can be used for decision making and adaptation to hydrological extremes (e.g., drought) especially for

agricultural sectors. And the Yellow River is a major farmland in China with intensive irrigations, where a dynamical-model-based seasonal hydrological forecasting system that is targeted for adaptation is quite necessary.

The introduction of the climate prediction part of the system and the evaluation of the NMME-based seasonal hydrological hindcasts during 1982-2010 will be presented in the second companion paper. In this paper, the establishment of the hydrological part of the forecast system (Fig. 3) is described below. The hydrological modeling part consists of the Variable

Infiltration Capacity (VIC; Liang et al. 1996) land surface hydrological model and a global routing model (Yuan et al., 2015a) regionalized over the Yellow River. The VIC model version 4.0.5 is used to predict soil moisture and runoff in this study. It is a semidistributed, grid-based hydrological model with a mosaic representation of land cover and soil water storage capacity. The VIC model is widely used to simulate the large-scale hydrology in China (Xie et al., 2007; Zhang et al., 2014).

The routing model, which is based on an aggregated network-response-function routing algorithm (Gong et al., 2009), uses the topographic data to calculate flow velocities both in the hillslopes and the channels, and translates the runoff from the VIC model to streamflow at each grid cell and routes the flow into rivers and finally into the ocean (Yuan et al., 2015a). Calibration of the VIC model and the routing model is described in section 2.3.

Figure 3 also shows that there is a hydrological post-processing part after the routing, which is necessary because there are

model uncertainties that cannot be calibrated (e.g., irrigation and inter-basin water diversion that are neglected in most large-scale land surface hydrological models) and the errors in meteorological forcings from climate forecast models can propagate nonlinearly after the terrestrial hydrological processes (Yuan and Wood, 2012). The hydrological post-processing will be used in the second companion paper by matching the predicted streamflow with observed streamflow, while in this paper the calibration and predictability assessment are based on the naturalized and simulated streamflow respectively. In

other words, this paper will assess the role of ICs in seasonal hydrological forecasting by neglecting the errors in calibrated hydrological models, and investigate the hydrological predictability in a "naturalized" Yellow River without human interventions. Assessing the naturalized hydrological predictability is the first step toward establishing an operational hydrological forecasting system, and will also provide a guideline for water resources management over the Yellow River basin.

**2.3 Calibration with naturalized streamflow**

The Yellow River basin is a heavily managed and intervened basin. As reported by the Bulletin of Water Resources, the observed annual mean streamflow at the outlet of the basin (i.e., Lijin station) is about $3.15 \times 10^{10}$ m$^3$ during 1956-2000, while the annual mean consumed and inter-basin diverted water is $1.48 \times 10^{10}$ m$^3$. These consumed and diverted water is usually neglected in the large-scale land surface hydrological models, and accounting for them in the model remains a grand

challenge due to limited water resources management data. In the second companion paper, the observed streamflow is used to correct the model simulations and forecasts for each target month through the post-processing techniques. While in this paper, the naturalized streamflow is used to calibrate the hydrological model and to investigate the naturalized or unperturbed hydrological predictability in terms of ICs. The naturalized streamflow is calculated by using the observed streamflow, the water consumed by agricultural, industrial and civil sectors, and the water regulated by reservoirs. In this

study, the naturalized streamflow datasets are obtained from the Bulletin of Water Resources.

The naturalized streamflow data at 12 gauges (Fig. 1) along the main stream of Yellow River and the rainfall data averaged over the sub-basins are used to calculate the runoff-rainfall ratios, and the grid-scale runoff time series over each sub-basin are then obtained by multiplying the runoff-rainfall ratios with rainfall time series. For the lower reaches, the difference in streamflow between the target gauge and the upstream gauge is used to calculate the runoff-rainfall ratios, with the rainfall

selected for those drainage areas between the two gauges. With the spatially disaggregated runoff time series, the parameters of the VIC model are calibrated automatically for each grid cell by using the Shuffled Complex Evolution (SCE) algorithm (Duan et al., 1994). The VIC model is run from 1951 to 1981 thousands of times, with the parameters searched by the SCE algorithm to obtain a maximum Nash-Sutcliffe efficiency (NSE) calculated between simulated runoff and naturalized runoff

during the period of 1961-1981, where the simulations in the first ten years (1951-1960) are dropped for spin-up. Similar automatic calibration procedure for the routing model is also carried out. It should be noted that the naturalized streamflow may contain errors from the measurement of precipitation and/or streamflow, and the errors may result in uncertainty in the calibrated parameters and the hydrological model. In the future, multisource (e.g., satellite and ground) observations combined with data assimilation techniques are needed to quantify such uncertainty.

Similar to Troy et al. (2008), seven parameters of the VIC model including the variable infiltration curve parameter ($b$, with the allowed range of 0.001-1), maximum baseflow velocity (Dsmax, 0.1-50 mm/day), fraction of Dsmax where nonlinear baseflow begins (Ds, 0.001-0.99), fraction of maximum soil moisture content above which nonlinear baseflow occurs (Ws, 0.2-0.99 and Ws>Ds), depths of the second and the third soil layers ($d_2$, $d_3$, with the range of 0.1-3 m; note that the depth of the first layer is fixed at 0.1 m) and the parameter characterizing the variation of saturated hydraulic conductivity with soil

moisture (the allowed range is 3.1-50), are selected for calibration. After the calibration of the VIC model at 1321 grid cells over the Yellow River, and simulated runoff with the optimized VIC parameters is used as the input for the routing model, and the flow velocities over the hillslope and within the channel are selected for calibration, with the allowed range of 0-1.0 m/s and 1.0-3.0 m/s respectively.

Table 1 lists the NSE calculated by using monthly naturalized and simulated streamflow during the calibration and validation

periods, and Figure 4 shows the time series of the streamflow at five selected gauges. A NSE value of one indicates that the model simulates the reference streamflow perfectly, and a value below zero indicates that the simulated streamflow is worse than the climatology. Across 12 gauges, the averaged NSE values during the calibration and validation periods are 0.86 and 0.82, with a range of 0.78-0.92 and 0.71-0.91 respectively (Table 1). This indicates that the calibrated hydrological simulation system captures the variations of the naturalized streamflow over the Yellow River basin quite well, which is also

shown in Figure 4. However, Figure 4 also shows that the modeling system underestimates the high flow at upper reaches (e.g., Tangnaihai) and overestimates the low flow at middle and lower reaches (e.g., Hekouzhen, Huayuankou, Lijin) of the Yellow River. The underestimation of high flow at upstreams might be due to the deficiency in the snow-melting module since the headwaters region is located in a cold and mountainous area, while the overestimation of low flow might be related to the uncertainties in the subsurface hydrological processes as well as the transport of surface water.

**2.4 Experimental design**

With the calibrated hydrological simulation system, a set of numerical experiments are carried out to investigate the role of the initial hydrological conditions (ICs) in the seasonal hydrological forecasting: (1) a continuous simulation from 1951 to 2010 is used to generate the ICs at the beginning of each calendar month and the reference streamflow and soil moisture for the assessment of the naturalized hydrological predictability over the Yellow River; (2) the Ensemble Streamflow Prediction

(ESP) simulations initialized at the beginning of each calendar month during 1982-2010, with ICs taken from the same date of experiment (1) and 28 realizations of 6-month meteorological forcings taken from the same period of the target year while excluding the target year. For example, for the ESP simulation starting from March 1983, the ICs are exactly the same as the experiment (1) on March 1983, and the 28 ensemble of meteorological forcings are those in the experiment (1) during the March-August of 1982, 1984, 1985, …, 2010, without using the forcings in the target year; (3) the reverse ESP (revESP) simulations similar to the experiment (2), with the simulations driven by the same meteorological forcings taken from the experiment (1) during the target year, but 28 ensemble of ICs taken from different years excluding the target year. For example, for the revESP simulation starting from March 1983, the meteorological forcings are those in the experiment (1) during the March-August of 1983, while the 28 ensemble of ICs are taken from the March of 1982, 1984, 1985, …, 2010, without using the ICs at March of the target year (i.e., 1983).

In this paper, all the analyses are based on the ensemble means of the realizations from ESP and revESP. The Root Mean Squared Error (RMSE) for ESP and revESP for each calendar month are calculated by using all 6-month simulations starting from the same calendar month during 1982-2010. And the RMSE ratio, which is defined as $RMSE_{ESP}/RMSE_{revESP}$, is used to assess whether the ICs or the meteorological forcings are more important in the prediction of streamflow and soil moisture. If the ratio is lower than one, the ICs prevail over the meteorological forcings in the predictability of the target hydrological variable (e.g., streamflow or soil moisture); and if the ratio is larger than one, then the meteorological forcings are more important.

3. **Results**

**3.1 Predictability of streamflow**

Figure 5 shows the RMSE ratio for different calendar months and lead times at twelve selected hydrological gauges from upstream to downstream of the Yellow River basin. For example, the blue line starting from January and ending at June in Figure 5a shows that the RMSE of streamflow from ESP simulation is lower than the revESP simulation in January and February, indicating that the ICs prevail over the meteorological forcings in the streamflow predictability during the first two months; the RMSE ratio is larger than one from April to June, which suggesting that the meteorological forcings are more important for the streamflow prediction after the first three months. In general, there is a gradual increase in the lead time where the ICs significantly contribute to the streamflow predictability (RMSE ratio less than one) from upstream to downstream gauges. From the Tangnaihai gauge to Shizuishan gauge, the influence of ICs could not persist for one month for the forecasts starting from spring or early summer (green lines in Figs. 5a-5f). However, from the Hekouzhen gauge down to Lijin gauge, the ICs significantly contribute to the streamflow predictability during the first month for all calendar months (Figs. 5g-5l).

From the gauge at the headwaters region to that at the outlet of the Yellow River basin, ICs significantly contribute to the streamflow predictability for up to 2-5 months for the forecasts initialized in fall and winter, and the meteorological forcings prevail over the ICs in the predictability after the first month for the forecasts initialized in spring and summer (Fig. 5). This indicates that the ICs have stronger control on the streamflow predictability during the dry seasons than that during the wet

seasons. An interesting feature is that ICs have the weakest control on the streamflow forecasts starting before the rainy season (May in Fig. 5), which suggests that the memory of the terrestrial hydrological system drops to the lowest level at the end of the dry season. This is similar to the results of the predictability of soil moisture and runoff over the river basins with strong seasonality (Shukla and Lettenmaier, 2011), where the ICs have the strongest and weakest control at the end of rainy season and dry season respectively.

For the ESP results shown in Figure 5, both the state variables for the surface water and subsurface water are set to the realistic values according to the continuous offline simulation driven by observed meteorological forcings. To distinguish the relative importance from different sources of water storage, an additional experiment is conducted by setting the surface water state in the routing model to that used in the revESP experiment, i.e., the ICs of surface water in the ESP experiment are replaced with the climatology values. The RMSE ratios of the ESP without the initialization of the surface water over that from the original revESP are then calculated similarly, and the results are shown in Figure 6.

The impact of the initialization of the routing model is less obvious in the headwaters region (e.g., Fig. 6a) given a smaller drainage area and a shorter travel time. When it goes to the downstream gauges, the RMSE ratios in the first month increase greatly. As compared with a full initialization (both the initializations of surface and subsurface water) in the ESP experiment (Fig. 5), the dominant role of ICs in the first month forecasts almost disappears for all calendar months (Figs. 6g-6l). Nevertheless, the RMSE ratios for the forecasts beyond the first month do not change, no matter for the upstream or downstream gauges (Fig. 6). This suggests that the memory from initial surface water lasts for less than a month over the Yellow River basin and would not affect the streamflow forecasting at long leads. However, it is the most important sources of predictability for the streamflow over a large river basin at a short time scale. The ICs of the surface water states are essential for a seamless hydrological forecasting system that aims at integrating short-term flooding forecast to seasonal drought prediction.

The above analyses are based on the full samples of the hindcasts. To investigate the role of ICs in the seasonal forecasts of hydrological extremes, the results conditional on the dry/wet conditions are investigated. Previous studies found that the ESP approach has low forecasting skill before the onset of the extreme events (Yuan et al., 2015a), but can be skillful during the recovery stage (Pan et al., 2013). Therefore, the forecasts with initial streamflow percentile (according to the continuous offline simulation) lower than 20% (or higher than 80%) are used to calculate the RMSE ratios, and the drought cases are shown in Figure 7. It is found that the RMSE ratios are increasing as compared with the results of the full samples (Fig. 5). The dominant role of ICs can persist for two months for the forecasts starting in some spring and summer months at the downstream gauge (Figs. 7l).

The orange lines in Figure 5 show that the RMSE ratios tend to converge at a specific target month after the rainy season, regardless of different forecast lead times. This is because the river basin is entering into the dry seasons where the variability of meteorological forcings becomes smaller. Such convergence becomes clearer during the drought periods (Fig. 7). Since the Yellow River has a strong seasonality in the hydro-climate, it is difficult to recover in a short time once the hydrological drought occurs at the end of the rainy season. In this case, the influence of ICs persists for a longer time, and

the RMSE ratios do not increase with the increase of the lead times (Fig. 7). This demonstrates the usefulness of the ESP approach that is mainly based on the information from ICs in forecasting the persistency of the hydrological droughts. In other words, the skill of seasonal climate prediction during the dry season is less important because the ICs dominant the hydrological predictability for a long time. The result for the wet cases (initial streamflow percentile larger than 80%) is similar, but the impact of ICs lasts for a longer time (not shown). This is reasonable because wetter ICs usually contain more memory, and the evaporation process that dries up the soil is a slower process. For the drier ICs, a single storm may damage all the prior information and the system becomes less predictable.

To conclude, Figure 8 shows the Maximum Lead Times (MLTs) where the ICs prevail over the meteorological forcings in the streamflow predictability along the main stream and major tributaries of the Yellow River. At the outlet of the Yellow River, the MLT is less than 2 months during March-September (Figs. 8c-8i) and longer than 5 months during October-November (Figs. 8j-8k), then drops to 4, 3, 2 months for the forecasts starting from December, January and February respectively (Figs. 8l, 8a-8b). This is consistent with the results from a global seasonal streamflow forecasting at a large river basin scale (Yossef et al., 2013).

Moreover, given that the hydrological forecasting system established in this study can route the runoff and calculate the streamflow grid by grid, Figure 8 also shows the variability of MLTs over the upstreams and tributaries. They generally follow the seasonality pattern of MLT at the outlet, with longer and shorter values during dry and wet seasons respectively. For the forecasts starting from November, the MLTs are beyond 5 months except for a part of the main course in the upstream of the Tangnaihai gauge, and the main course between the Huayankou and Gaocun gauges (Fig. 8k). While for the forecasts starting from May, the MLTs are less than one month except for the main course between the Hekouzhen and Sanmenxia gauges, and that from the Gaocun gauge down to the outlet. Regardless the tributaries, the biggest difference in MLT between the lower reaches and upper reaches of the Yellow River occurs for the forecasts starting from October (the end of the rainy season), where the difference can be as large as 5 months (Fig. 8j). During the rainy season, the difference in MLT is about one month (Figs. 8f-8h).

### 3.2 Predictability of soil moisture

While the change of streamflow is mainly based on fast hydrological processes including the rainfall-runoff and runoff-routing processes, the change of soil moisture is much slower due to less conductivity of soil water. Therefore, the impact of ICs on the soil moisture forecasting is expected to be more significant than the streamflow. Figure 9 shows the same MLT plots as Figure 8, but for soil moisture. Similar to the streamflow (Fig. 8), the MLT for soil moisture is longer during the cold and dry seasons, and is shorter during the warm and rainy seasons (Fig. 9). However, unlike the streamflow that represents a basin-scale runoff variability where the lower reaches are connected with upper reaches, the grid-scale soil moisture only represents the local variability, and the soil moisture from upper to lower reaches of the Yellow River has no connections under the current hydrological modeling framework. In other words, the MLT for the soil moisture in the lower reaches is not necessarily longer than that in the upper reaches. As a result, the MLTs for the forecasts starting from September-February are beyond 6 months in the middle reaches of the Yellow River due to a dry climate (Fig. 2c), while the

MLTs are about 3-5 months in the upper reaches up to the Lanzhou gauge and the lower reaches between the Longmen and Huayuankou gauges (Figs. 9a-9b, 9i-9l). This pattern holds for the warm seasons: the ICs prevail over the meteorological forcings in the soil moisture predictability over the middle reaches for up to 4 months for the forecasts starting from spring (Figs. 9c-9e) and up to 2-3 months for the summer, while the MLTs are less than 1-2 months over the upper and lower reaches during the same period (Figs. 9f-9h).

Similar to the RMSE ratio analysis during the dry period (Fig. 7), the differences in MLTs between the dry cases and the average results (Fig. 9) are shown in Figure 10. The soil moisture time series can be converted into percentiles to form a drought index that is important for the indication of agricultural drought. In this study, the soil moisture fields from the continuous VIC simulation driven by observed meteorological forcings are converted to percentiles grid by grid to identify the local agricultural drought periods. Again, the ESP and revESP forecasts starting from the dry years (but the ICs or meteorological forcings from the two experiments are not necessarily dry according to their experimental design) are used to compute the RMSE ratios as well as the MLTs.

Figure 10 shows that the MLTs increase by one month over most areas. For the forecasts starting in the summer and early autumn, the increases can reach two months over the middle reaches and part of the upper reaches (Figs. 10f-10i). The stronger persistency of the dry soil indicates that the investment on the seasonal drought forecasting should not neglect the improvement in the ICs. 1 or 2-month increase in the forecast lead time will greatly benefit the agricultural preparedness for the drought events. Given that the seasonal forecast skill for the precipitation is quite limited beyond one month (Wood et al., 2015), the refinement of ICs through data assimilation techniques would be very important for the drought forecasting, especially over the middle reaches of the Yellow River where several main farmlands exist. The MLTs over the middle reaches during the cold seasons remain the same because the original MLTs reach the 6-month limit (Fig. 9). In other words, they may also increase if the ESP and revESP experiments are carried out to the 7[th] months or forward. The increases in the MLTs for the wet cases are more significant (not shown), suggesting that wetter ICs could dominate the soil moisture predictability longer than drier ICs.

4. **Concluding Remarks**

This is the first paper of a two-part series on introducing an experimental seasonal hydrological forecasting system over the Yellow River basin in northern China. While the second companion paper will focus on the evaluation of the North American Multimodel Ensemble (NMME)-based seasonal hydrological forecasting (Yuan, 2015), this paper introduces the system and uses it to investigate the role of initial hydrological conditions (ICs) over the Yellow River basin.

The forecasting system is similar to the global forecasting system established by Yuan et al. (2015a), but with a higher resolution and a finer calibration procedure. Based on five decades (1961-2010) of the naturalized streamflow datasets at twelve mainstream gauges that are recently compiled by the Yellow River Conservancy Commission, as well as a new forcings dataset compiled from 324 meteorological stations, a land surface hydrological model and a global routing model regionalized over the Yellow River are calibrated grid by grid at a 0.25-degree resolution through an automatic calibration method. The spatially disaggregated calibration results in averaged Nash-Sutcliffe efficiency of 0.86 and 0.82 for the twelve

gauges during the calibration and validation periods, respectively. In addition, a hydrological post-processor is used to transfer the naturalized, simulated or predicted streamflow to those comparable to the observed streamflow, which is essential for an operational seasonal hydrological forecasting over the Yellow River where the irrigations and inter-basin water diversions occur extensively.

By using the hydrological part of the forecasting system, a set of Ensemble Streamflow Prediction (ESP) and reverse ESP-type simulations that consist of 12 (months) × 29 (years during 1982-2010) × 28 (ensembles) × 6 (forecast leads) × 2 (ESP and revESP) = 116,928 months model integrations over 1321 grid cells, are conducted to investigate the role of ICs in seasonal hydrological forecasting over the Yellow River. For the streamflow prediction at twelve mainstream gauges, there is a gradual increase in the lead time where the ICs prevail over the meteorological forcings in the predictability. ICs

outweigh the meteorological forcings up to 2-5 months during the cold and dry seasons, but the meteorological forcings prevail over the ICs in the streamflow predictability after the first month during the warm and wet seasons. And from the Tangnaihai gauge at the headwaters region down to the Shizuishan gauge at the middle reaches, the ICs have very limited role (less than a month) for the forecasts starting before the rainy season.

Given that the ICs of surface water might be an important source of streamflow predictability, an additional ESP-type

simulation is conducted by setting the ICs of surface water to the climatology. Compared with revESP simulation, it is found that the initial surface water state is the most important source of streamflow predictability during the first month, especially for the downstream areas. However, there is no significant difference in the streamflow forecasting beyond one month regardless of whether initializing the surface water state or not, suggesting that other sources of terrestrial memory such as the snow and soil water storage become more important for the long-term streamflow predictability.

The role of ICs could be more significant during the dry/wet periods, where the dominance on the streamflow predictability at the lower gauges can be extended by a month even in the rainy season. This indicates that the ESP is a useful hydrological forecasting method after the onsets of the hydrological droughts or wet spells. The Maximum Lead Times (MLTs) where the ICs prevail over the meteorological forcings in the streamflow predictability at the outlet of the entire Yellow River are about 1 month and 5 months for the forecasts initialized during March-September and October-November respectively, and

increase from 2 months to 4 months for the forecasts initialized between them. There is a 5-month difference in MLT between the lower reaches and upper reaches of the Yellow River for the forecasts initialized at the end of the rainy season, while there is only 1-month difference during the rainy season.

Similar analysis is applied for the soil moisture, where the MLT for soil moisture is generally higher than the streamflow. The MLTs for soil moisture in the middle reaches of the Yellow River are about 6 months during the dry seasons, and they

drop to 2-5 months for the upper and lower reaches. However, the memory of soil moisture needs to be assessed more objectively by using in-situ and remote sensing observations because currently only the streamflow observations are used to constrain the hydrological models, where the soil moisture in the model can only be corrected implicitly based on the water balance equations.

Although this study have assessed the natural hydrological predictability that is important for an operational hydrological forecasting with water allocations and abstractions over the Yellow River, there are a few concerns that should be addressed in the future: 1) a multimodel framework (Koster et al., 2010) is necessary to quantify the uncertainty for the assessment of hydrological predictability; 2) the revESP method only assesses the theoretical predictability control by using all historical

ICs. Actually, operational forecaster can refine the ICs to some extent before issuing the forecasts because of the tendency in the ICs (i.e., prior information). In this regard, the revESP may overestimate the uncertainty in the ICs. On the other hand, the ESP method may also overestimate the uncertainty in the meteorological forcings because a conditional ESP method that is based on certain teleconnections (van Dijk et al., 2013) can be used to select the meteorological forcings from all historical samples. A more elastic method that is recently proposed by Wood et al. (2016) could be used to understand the role of ICs

in the seasonal hydrological forecasting with various level of uncertainty; 3) the hydrological predictability cannot be fully understood without combining the hydrological modeling approach and observation dataset to address different sources of predictability arising from surface water, soil water and/or groundwater, and the satellite retrievals of stream stage, soil moisture and terrestrial water storage would be important for the predictability studies over a large river basin; and 4) for the river basins with intensive water resources management, understanding of the naturalized hydrological predictability is just a

first step, more efforts should be devoted to improving the understanding of a "real" hydrological predictability by incorporating human interventions. This is also along the line with the Panta Rhei project that is proposed by the International Association of Hydrological Sciences in 2013, to understand, predict and manage the water systems that are increasingly impacted by humans, and to provide support for the adaptation to a changing environment.

**Acknowledgement.** This work was supported by the National Natural Science Foundation of China (No. 91547103), and the Thousand Talents Program for Distinguished Young Scholars. We would like to thank Dr. V. Moreydo and an anonymous reviewer for their helpful comments, and thank Joshua Roundy for the implementation of the routing model.

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

**Table 1.** Information at twelve hydrological gauges and the Nash-Sutcliffe efficiency (NSE) during the periods of calibration (1961-1981) and validation (1982-2010). The simulated streamflow is verified against naturalized streamflow.

| Gauge | Latitude (°N) | Longitude (°E) | Drainage Area ($10^3$ km$^2$) | NSE for Calibration | NSE for Validation |
|---|---|---|---|---|---|
| Tangnaihai | 35.5 | 100.15 | 122 | 0.90 | 0.87 |
| Xunhua | 35.83 | 102.5 | 145 | 0.91 | 0.88 |
| Xiaochuan | 35.93 | 103.03 | 182 | 0.78 | 0.84 |
| Lanzhou | 36.07 | 103.82 | 223 | 0.92 | 0.91 |
| Xiaheyan | 37.45 | 105.05 | 254 | 0.92 | 0.90 |
| Shizuishan | 39.25 | 106.78 | 309 | 0.92 | 0.89 |
| Hekouzhen | 40.25 | 111.17 | 368 | 0.86 | 0.76 |
| Longmen | 35.67 | 110.58 | 498 | 0.83 | 0.74 |
| Sanmenxia | 34.82 | 111.37 | 688 | 0.83 | 0.77 |
| Huayuankou | 34.92 | 113.65 | 730 | 0.85 | 0.81 |
| Gaocun | 35.38 | 115.08 | 734 | 0.84 | 0.78 |
| Lijin | 37.52 | 118.3 | 752 | 0.79 | 0.71 |

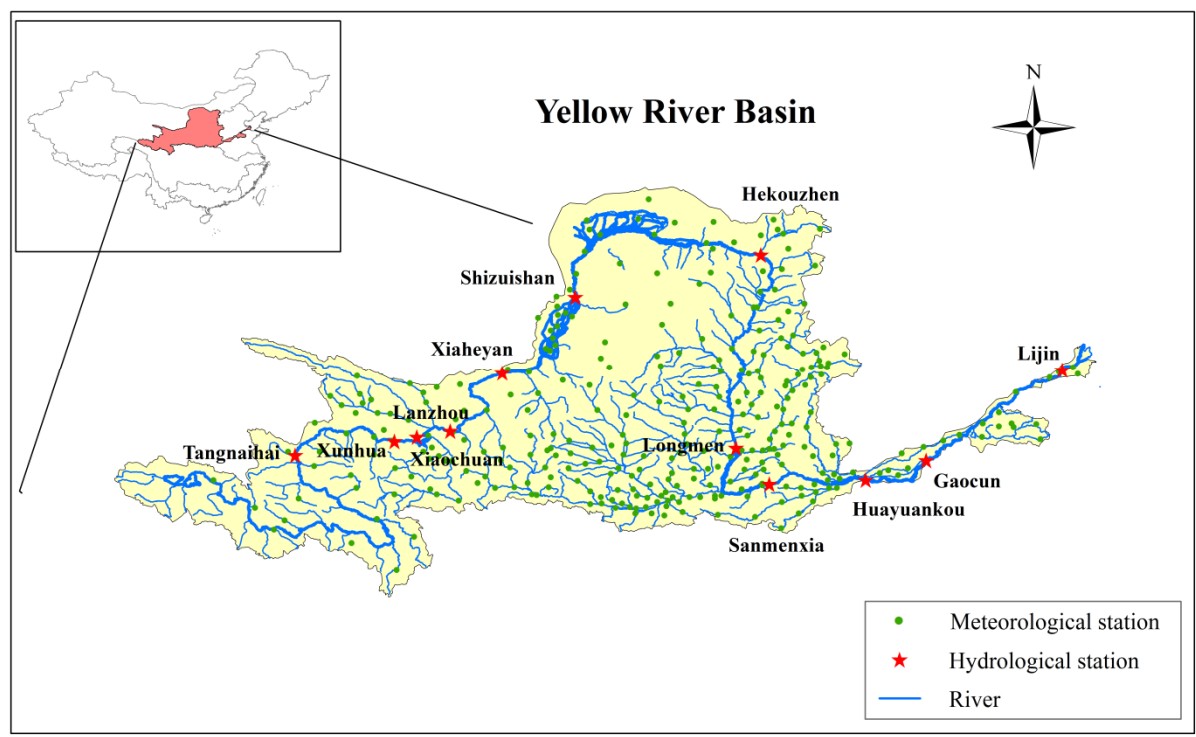

**Figure 1.** Locations of meteorological and hydrological stations over the Yellow River basin.

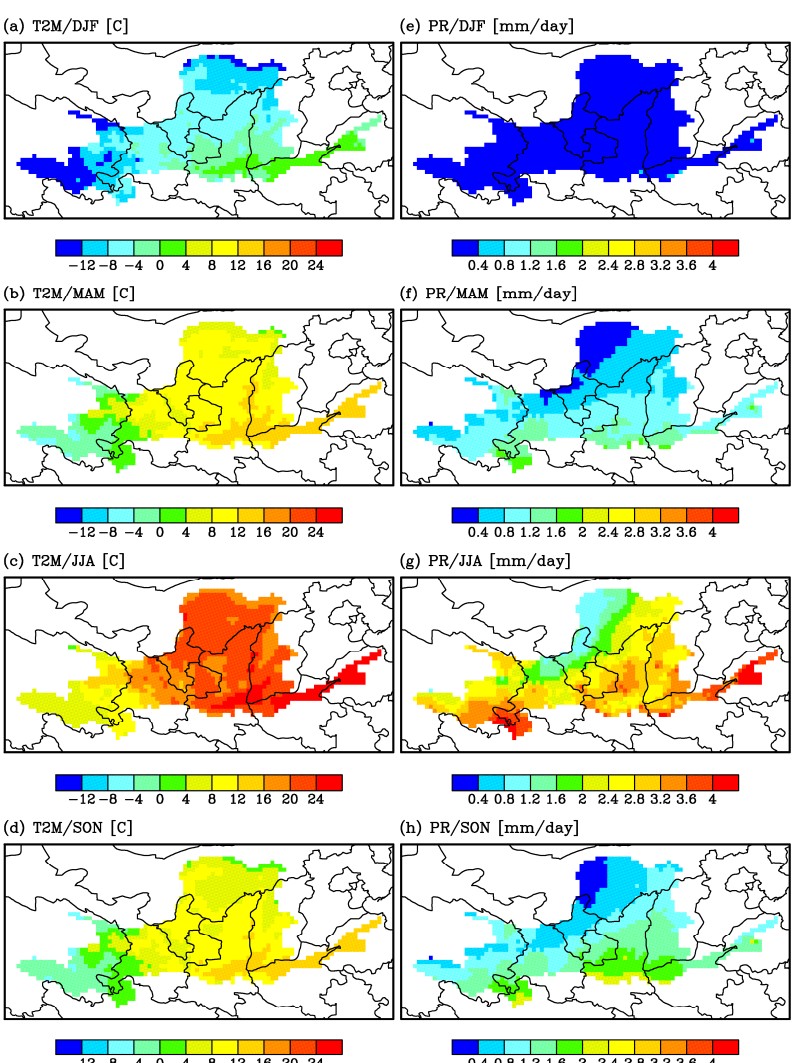

**Figure 2.** Seasonal mean (a-d) 2-m air temperature and (e-h) precipitation over the Yellow River averaged during 1982-2014. The four seasons are December-January-February (DJF), March-April-May (MAM), June-July-August (JJA) and September-October-November (SON).

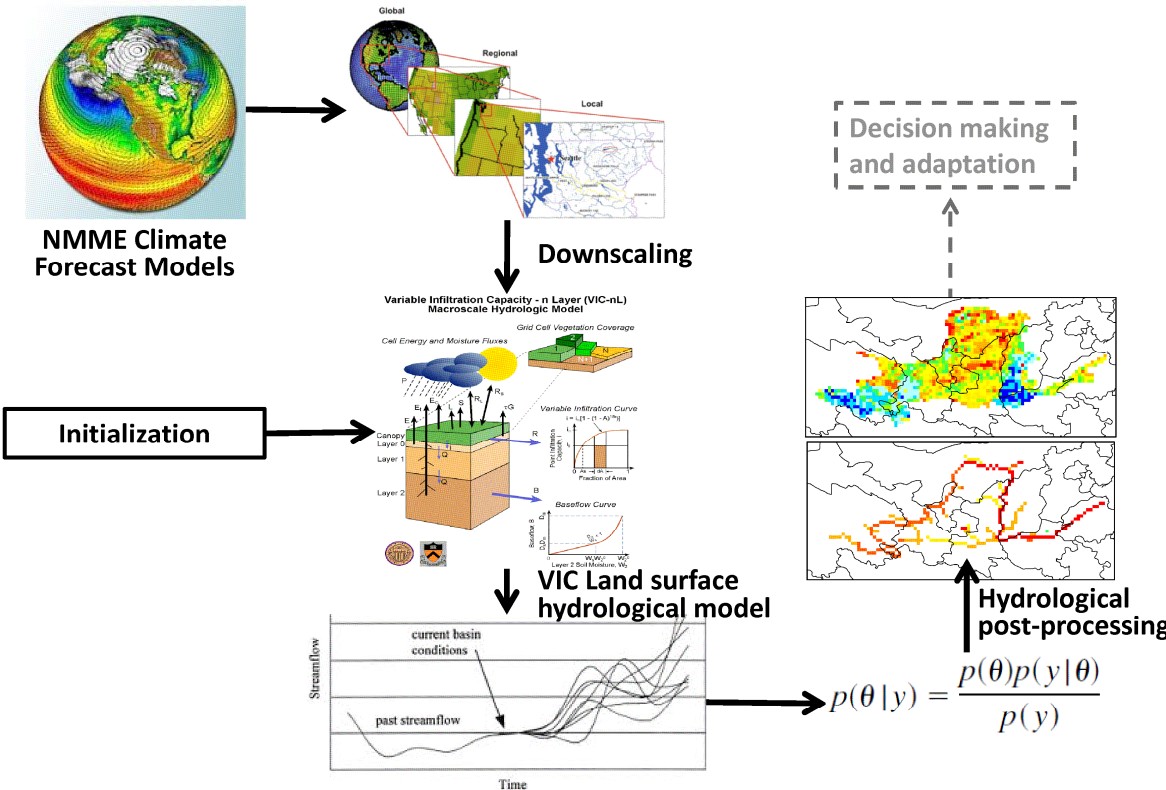

**Figure 3.** Flowchart for the experimental seasonal hydrological forecasting system over the Yellow River.

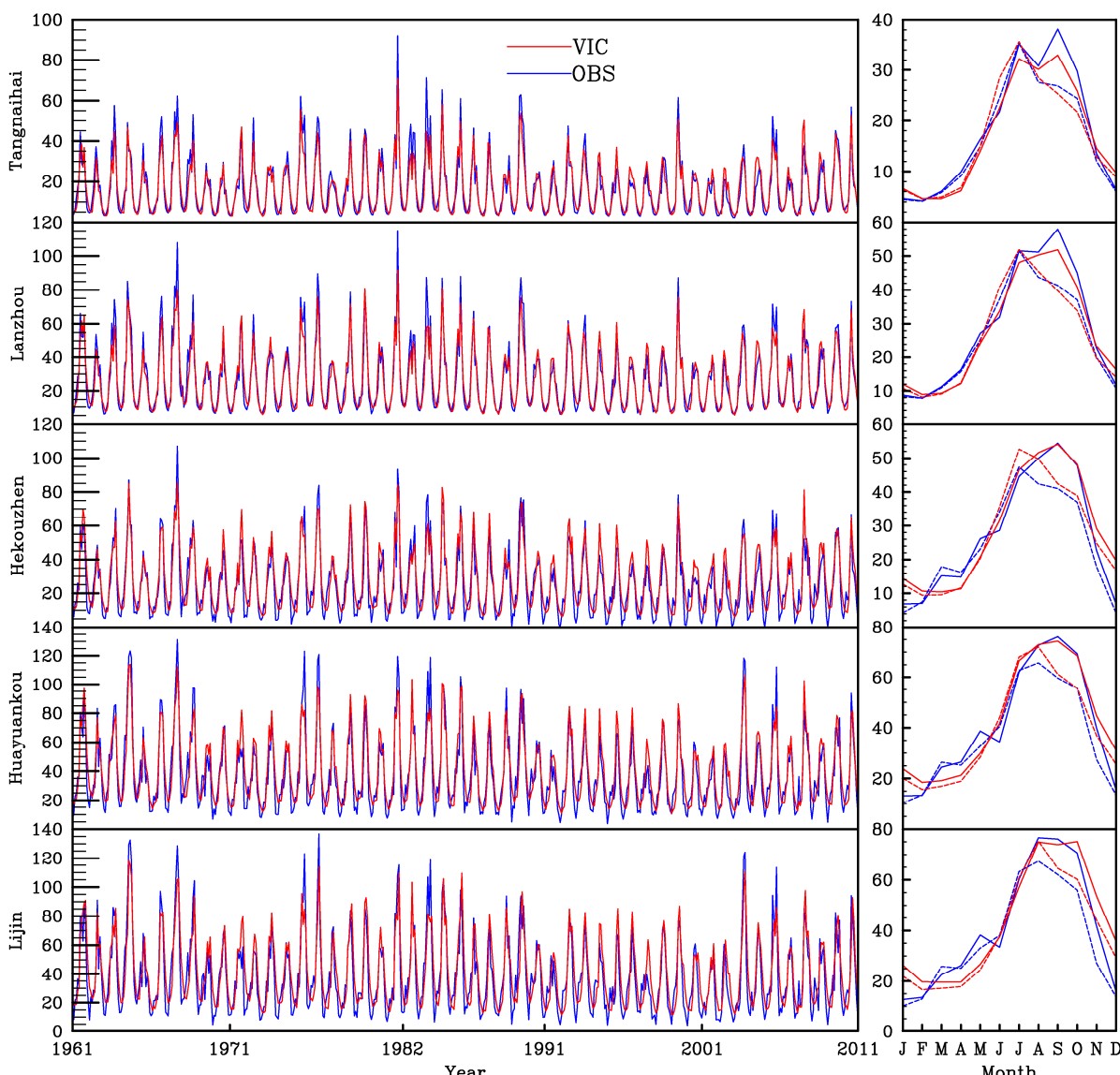

**Figure 4.** Naturalized (blue) and VIC simulated (red) monthly streamflow ($10^8$ m$^3$/s) at five hydrological gauges located from upper to lower mainstream of the Yellow River. The solid and dashed lines in the right panels represent the climatologies during the calibration (1961-1981) and validation (1982-2010) periods.

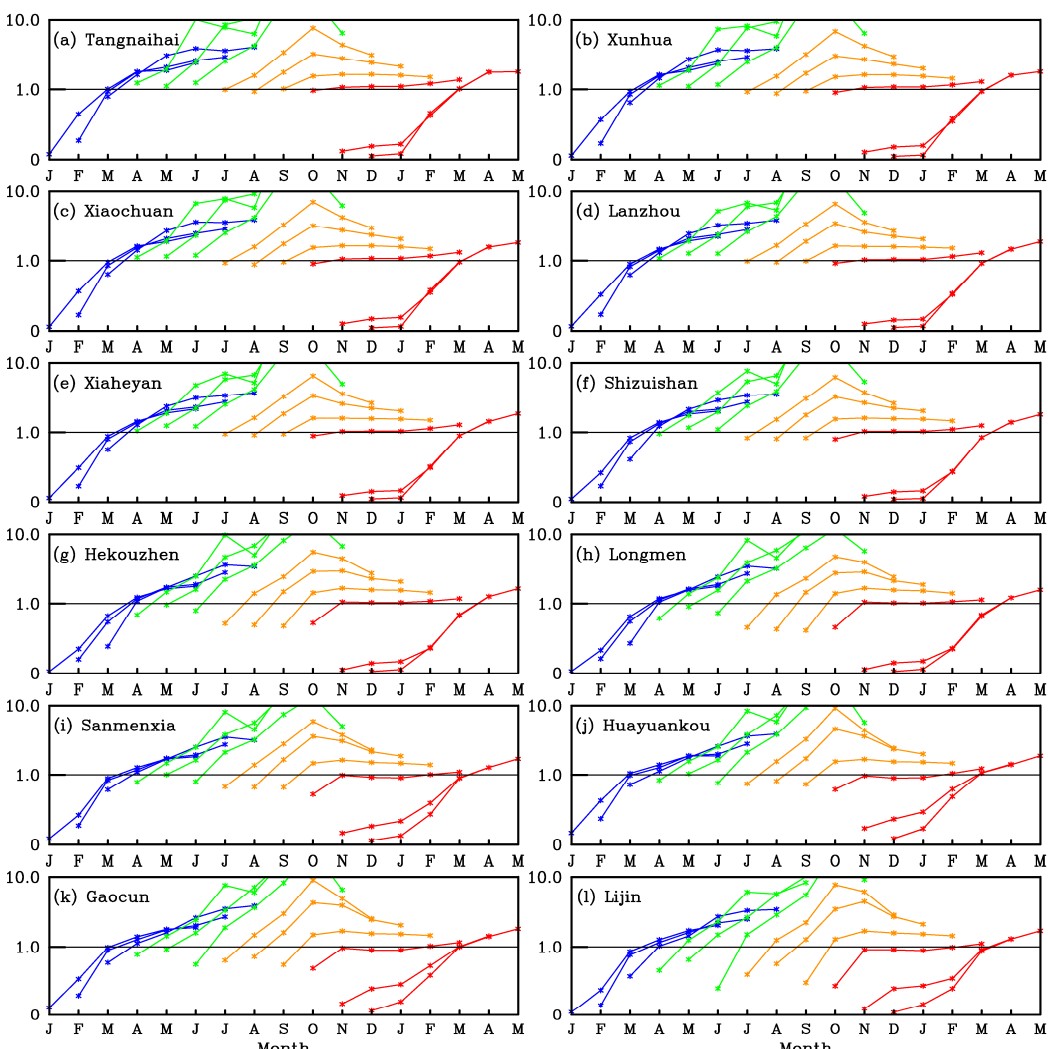

**Figure 5.** The Root Mean Squared Error (RMSE) ratio (RMSE$_{ESP}$/RMSE$_{revESP}$) as a function of start month and lead time at twelve hydrological gauges. The RMSE is calculated between the streamflow from a continuous simulation (with accurate initial condition and meteorological forcing) and that from the ESP or revESP experiments.

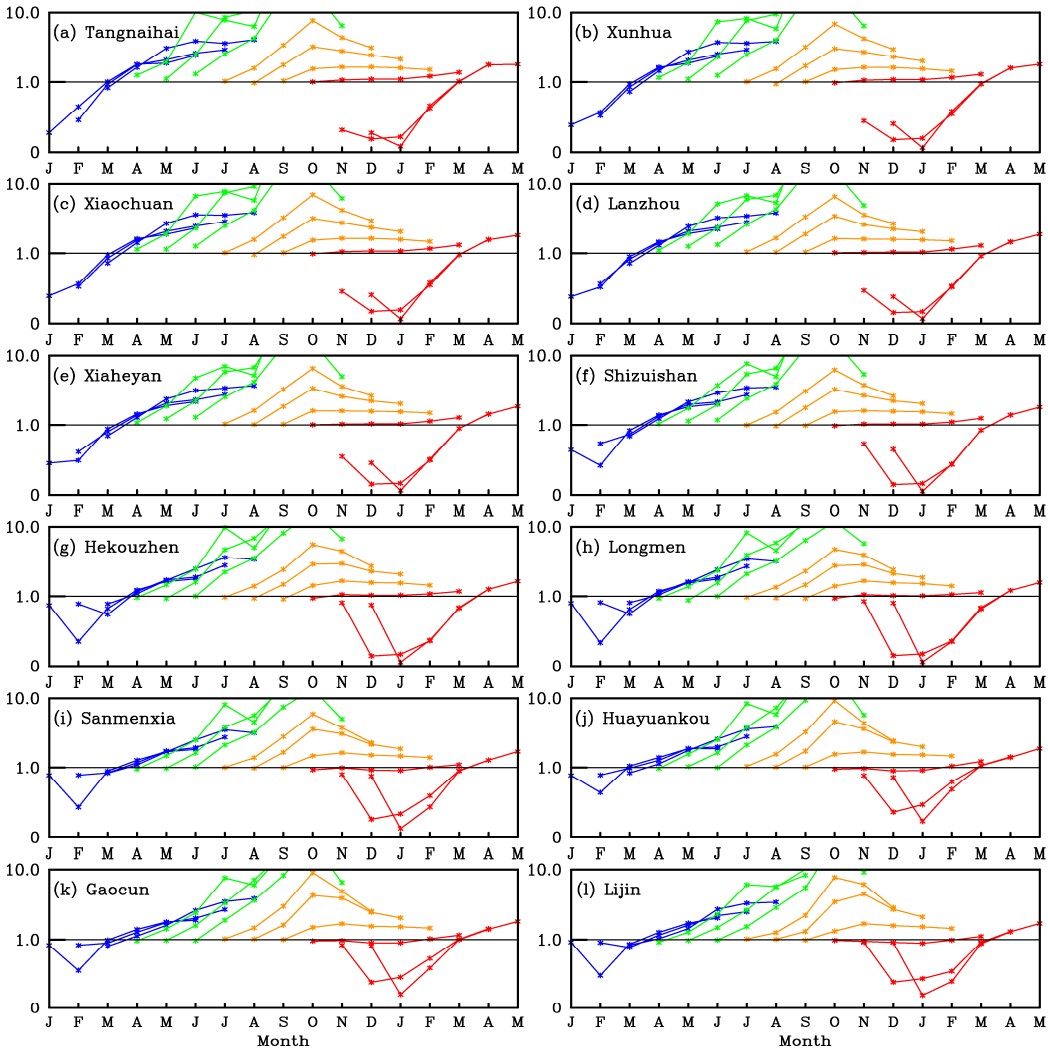

**Figure 6.** The same as Fig. 5, but for the ESP simulations without the initialization of the routing model.

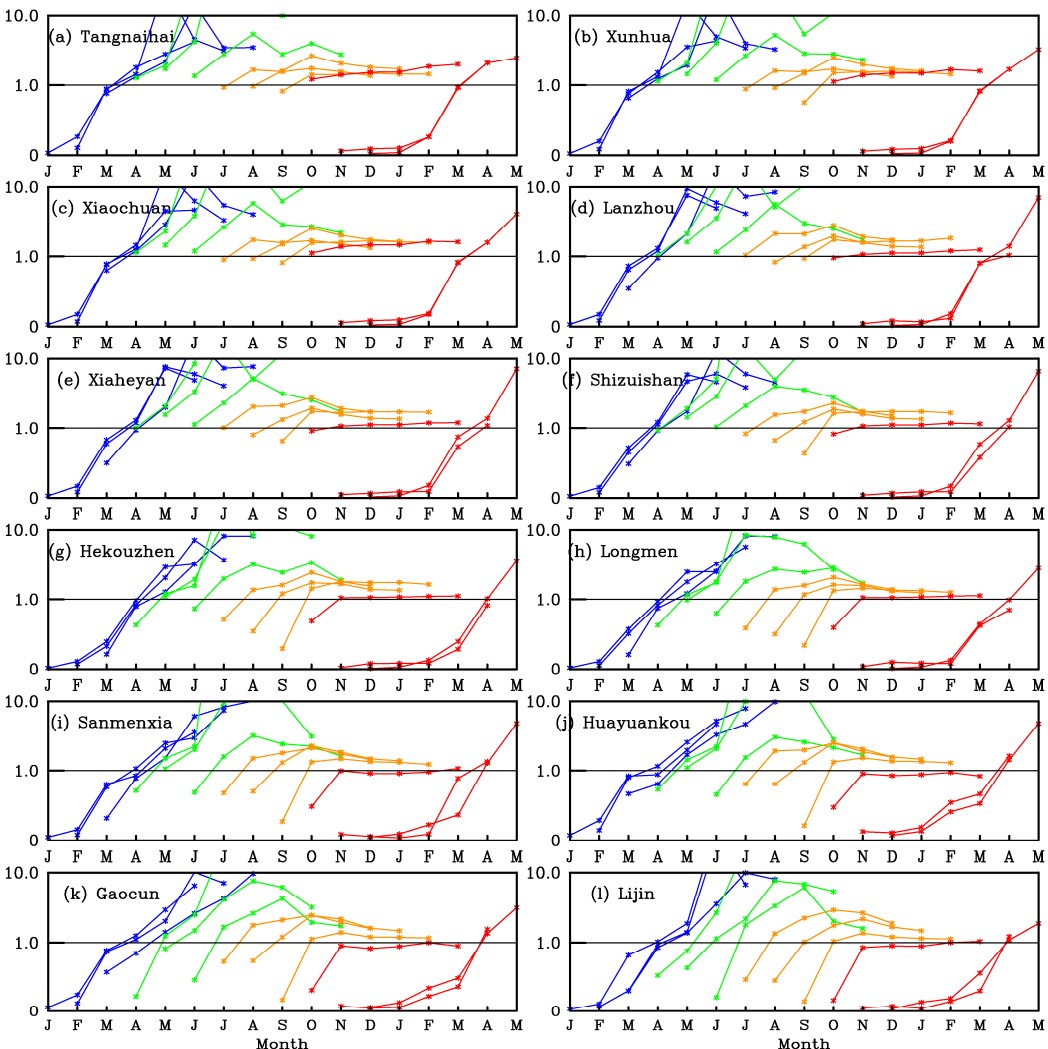

**Figure 7.** The same as Fig. 5, but for those years with streamflow percentiles at the start month lower than the 20%.

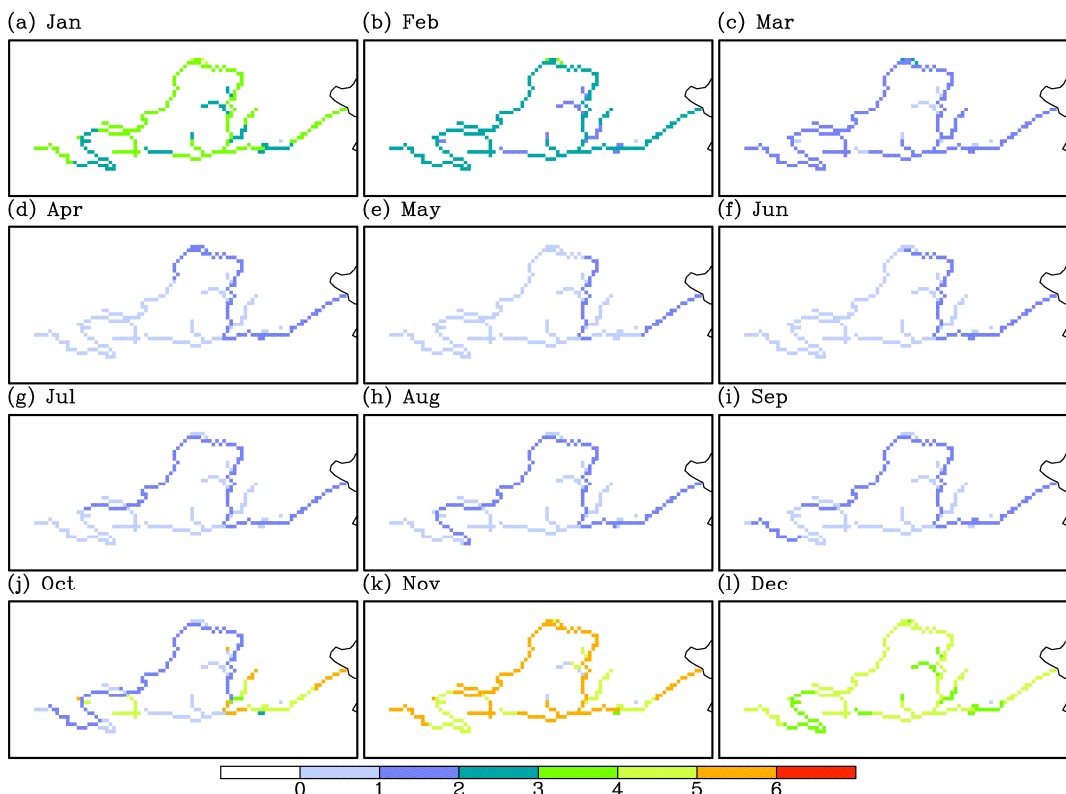

**Figure 8.** Maximum lead time (months) where the initial conditions prevail over the meteorological forcings $(\mathrm{RMSE}_{\mathrm{ESP}}/\mathrm{RMSE}_{\mathrm{revESP}}<1)$ in the streamflow predictability.

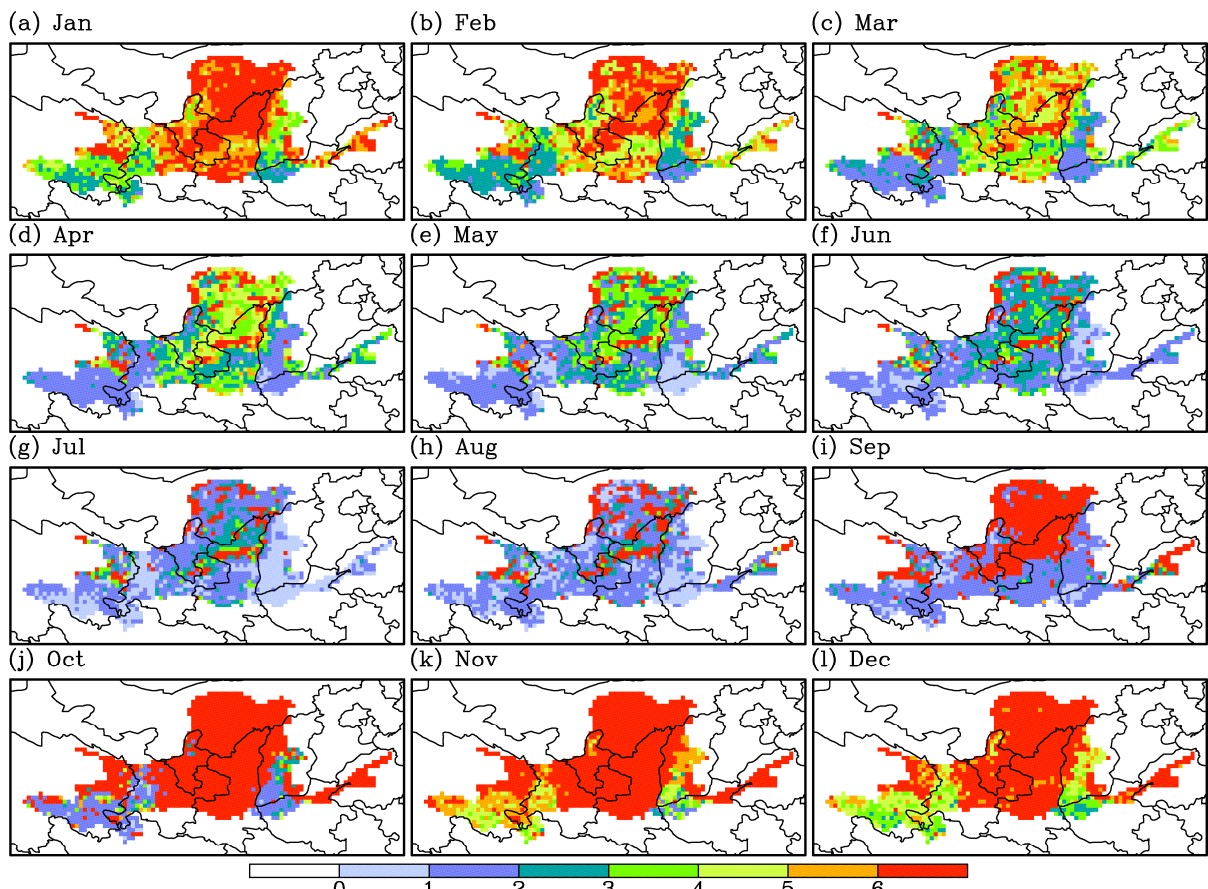

**Figure 9.** The same as Fig. 8, but for soil moisture.

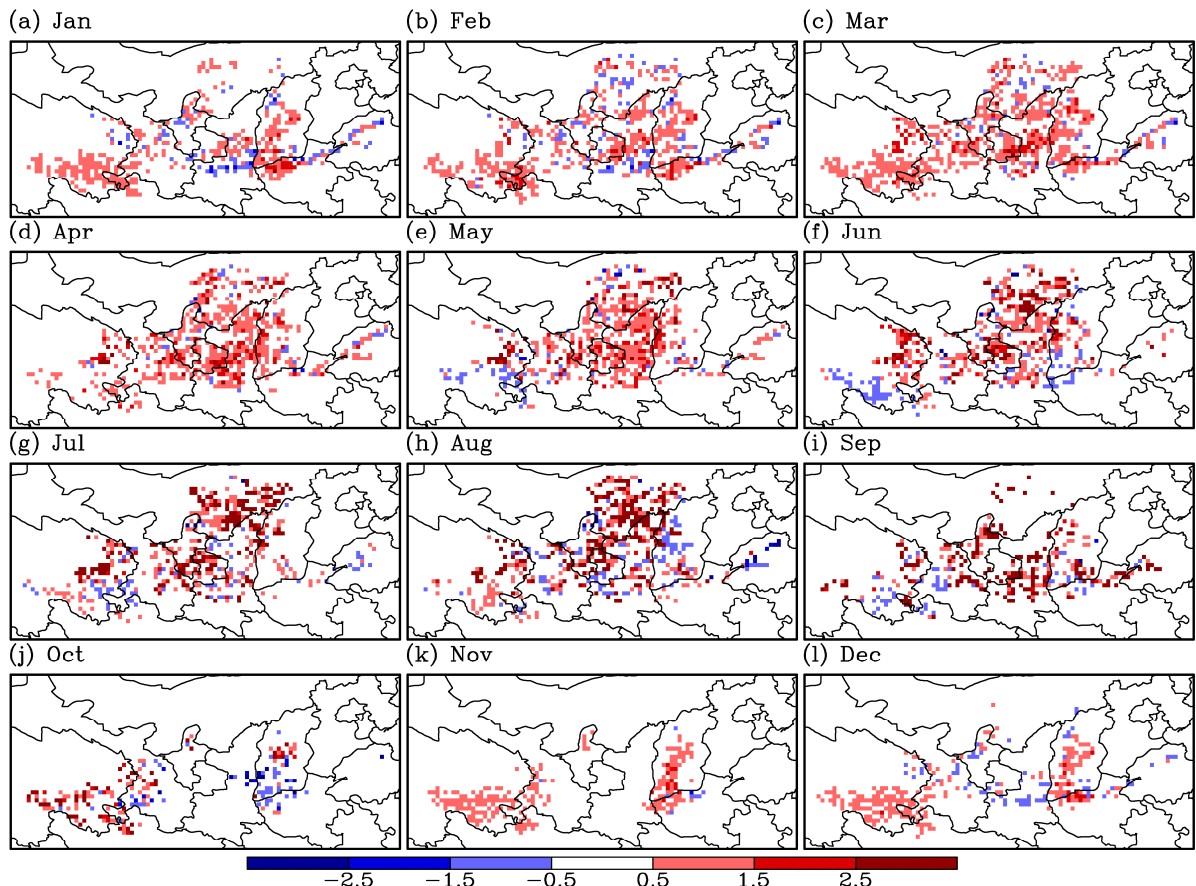

**Figure 10.** Differences in maximum lead times (months) between dry years (with soil moisture percentile lower than 20%) and the mean results for soil moisture.