# Peer review of "An experimental seasonal hydrological forecasting system over the Yellow River basin-Part I: Understanding the role of initial hydrological conditions"

_Hydrology and Earth System Sciences, 2016_

## Referee Comment (RC1) · V. Moreydo (Referee) · 2 Apr 2016

A set of two papers by Xing Yuan is presented for the review. The first one is devoted to the investigation of the role of initial watershed conditions for streamflow forecasting in the Yellow River basin, China. The second paper deals with seasonal ensemble forecasting of the human-induced streamflow by the means of North American Multimodel Ensemble (NMME) and forecast post-processing procedure. Given a vast material and several main issues under consideration within a single modeling framework, splitting up the papers is a good decision. This comment addresses the first paper of the two.

The paper investigates the predictability of streamflow and soil moisture on a seasonal extent (up to 6 months ahead) within a large river basin, characterized with different streamflow formation patterns. Seasonal streamflow predictability studies are considered to be one of the major directions for the hydrological scientific community (Blöschl, 2006), and the presented paper is aimed to outline the regional patterns of predictability, thus fully corresponding with the international hydrological agenda. The authors assess contributions of initial watershed conditions and meteorological forcings within the forecast window to seasonal predictability of the streamflow along the main course of the Yellow River. To fulfill the task, an experiment is designed based on the Reverse-Ensemble streamflow prediction method (rev-ESP, Wood and Lettenmaier, 2008). The Variable infiltration capacity (VIC) hydrological model (Liang et al., 1996) with global routing model by Yang et al. (2015) is used to describe the initial conditions and simulate the streamflow for the test period of 1982 – 2010. Meteorological forcings are set on a grid-base with 0.25 degree resolution holding the interpolated data from 324 meteorological stations within the basin. The hydrological model is calibrated against naturalized streamflow (reconstructed for the whole basin with the rainfall data using rainfall-runoff ratio at 12 main river gauges) and shows overall good performance with NSE values of 0.78 – 0.91. One of the issues the reviewer would like to address concerning the experiment is that the naturalized streamflow may contain errors concerning the precipitation and streamflow measurements. Reaching up to 10% of the measured value, the errors, when combined, might introduce uncertainties in the naturalized streamflow time-series. The uncertainties may be transferred forward to the hydrological model by calibrating it against these time-series which may result in unrealistic parameters. It should be noted that the authors use naturalized streamflow in order to assess natural and unperturbed hydrological predictability apart of the human induced water subtractions. The observed streamflow is used in the second paper to condition the hydrological forecasts. The overall design of the experiment complies with the ESP/revESP framework. For the ESP experiments the model is first run from 1951 to 2010 to simulate the natural initial conditions of the basin, then for the ESP

experiment the simulation is initialized at the beginning of each calendar month within the period 1982 – 2010 with 28 6-month-long realizations of meteorological forcing. For the revESP experiment the model is run again for 6 months starting each month within the test period with single actual meteorological forcing, but the initial conditions are taken from the ensemble of 28 members excluding the year under consideration. The memory of the hydrological system is demonstrated by calculating the ratio of the RMSE from ESP and revESP experiments, respectively. The ratio values less than 1 show initial conditions prevailing over meteorological forcing in streamflow predictability over the forecast period and the memory of the hydrological system is controlling the streamflow conditions. The ratio greater than 1 shows that the predictability depends on climatic system and the memory of the hydrological system has no control over the streamflow. Results obtained for 12 gauges along the river course show that the initial conditions control the system in Autumn and Winter, while meteorological conditions tend to influence the system in the Spring and Summer rainy season. The authors outline that it has been discovered that the hydrological system memory drops to the lowest level at the end of the dry season. Further investigation of the initial surface water conditions is also carried out, concluding that the system memory of the surface water state lasts for less than a month in general and has no influence on the long lead-time, yet for the short leads it is crucial. Next, the initial conditions are selected to attribute for a wet and dry state of the basin, thus tending the model to reproduce extremes. This part concludes that the wet state of the basin is generally more influenced by the initial conditions, while the drier case is less stable. To assess the predictability of the soil moisture the same ESP/revESP technique is applied, showing that again the best forecast performance for such a slow process as soil moisture transport is obtained for the dry winter season and a significant lead-time up to 2 – 3 months can be still obtained for summer months. The overall characteristic of the paper is very good, it is a major contribution to global investigations of the limits of hydrological predictability. The paper can be published without any revisions.

---

## Referee Comment (RC2) · Anonymous Referee #2 · 13 Apr 2016

This manuscript presents a seasonal hydrological forecast system for the Yellow River basin and investigates the contribution of hydrological initial condition and meteorological forcing to the predictability of soil moisture and streamflow over the study region. The topic is suitable for HESS, and the research method is scientifically sound. The manuscript is generally well written with good quality illustrations. It is a good piece of work. But I am a little disappointed with the scientific contribution of this work to our knowledge and understanding about seasonal hydrological forecasting, at least in the way that was presented. This is one of my major concerns. There are also a number of places in the manuscript that needs clarification or justification. Overall, I think a major

revision is necessary to improve the quality of the paper.

Major concerns This is a solid piece of research work, but there isn't really anything new in terms of research methods or scientific understanding. Several previous studies have adopted the exact same methodology and answered the exact same questions, but just over different basins. So besides applying the same methodology in a forecast system over the Yellow River basin (new to some degree), there is not enough evidence to support the novelty of this research. The authors argue that a new meteorological dataset with higher resolution is used, but it was not demonstrated how this improved resolution actually help with the hydrological forecasting. My other concern is on the revESP approach as a way to estimate the impact of IC uncertainties on hydrological forecasting. Although this approach has been used in several published studies, it is still necessary to point out that this approach significantly overestimates the uncertainty associated with IC as it uses all historical ICs. This is more so than the ESP approach for meteorological forcing. Please note that the meteorological forcing is during the forecast period which is unknown at the time of forecast, but the IC is just not able to be completely observed. The IC is the result of past meteorological conditions that have been observed to a large degree. So cautions need to be raised when interpretation of the results (ESP vs revESP), and some discussion is necessary on this issue in the end. Minor issues 1ïïjŐ Page 2 line 9: what is a more extreme climate? 2ïïjŐ Page 2 line 11: Some references are needed to back up this statement. 3ïïjŐ Page 2 line 12: Some references are needed here, too. 4ïïjŐ Page 2 line 14: There is a different between mitigation and adaption. Should seasonal forecast be more helpful with mitigation instead of adaptation? Adaptation usually happens at much longer time scales. 5ïïjŐ Page 2 line 19: Why Atlantic Ocean? What about other oceans? 6ïïjŐ Page 4, line 10: regridding usually means changing the spatial resolution of a grid data product. Here the station data is interpolated somehow to a fixed grid, so it is not regridding. It is also necessary to mention how the interpolation is done. 7ïïjŐ Page 4 line 20: "river is suspended ??" What do you mean by that? I guess what you want to say is that "the riverbed is elevated above the adjacent floodplains due to

sediment deposition and man-made levees". 8ïijŐ Page 5 line 20-21: Do you have a source for these statistics? 9ïijŐ Page 7 line 17: "dominant role of IC's for streamflow predictability". See the major concern #2. This interpretation needs to be cautious. 10ïijŐ Page 8, line 7: what is a full initialization? 11ïijŐ Figure 2: this is useful to show the spatial variation of mean temperature, precipitation and wind. But it is not the most useful ones, for example the wind is never discussed in the study. It is actually necessary to show the seasonal cycle of precipitation (and probably temperature) over the basin, just because you use such information in Figure 8. 12ïijŐ Figure 8: why are there a number of small streams showing the max lead time of 6 months all the time?

---

## Author Comment (AC2) · 23 Apr 2016

We are very grateful to the reviewer for the positive and careful review. The thoughtful comments have helped improve the manuscript. The reviewer's comments are italicized and our responses immediately follow.

This manuscript presents a seasonal hydrological forecast system for the Yellow River basin and investigates the contribution of hydrological initial condition and meteorological forcing to the predictability of soil moisture and streamflow over the study region. The topic is suitable for HESS, and the research method is scientifically sound. The

manuscript is generally well written with good quality illustrations. It is a good piece of work. But I am a little disappointed with the scientific contribution of this work to our knowledge and understanding about seasonal hydrological forecasting, at least in the way that was presented. This is one of my major concerns. There are also a number of places in the manuscript that needs clarification or justification. Overall, I think a major revision is necessary to improve the quality of the paper.

Response: We thank the reviewer for the comments. Please see our clarification of the novelty and the responses to the comments below.

Major concerns:

This is a solid piece of research work, but there isn't really anything new in terms of research methods or scientific understanding. Several previous studies have adopted the exact same methodology and answered the exact same questions, but just over different basins. So besides applying the same methodology in a forecast system over the Yellow River basin (new to some degree), there is not enough evidence to support the novelty of this research. The authors argue that a new meteorological dataset with higher resolution is used, but it was not demonstrated how this improved resolution actually help with the hydrological forecasting.

Response: We thank the reviewer for the comments. The two companion papers introduce an experimental seasonal hydrological forecasting system over the Yellow River basin in northern China to provide adaptive support in a changing environment. The system draws from a legacy of a global hydrological forecasting system (Yuan et al., 2015), but with the VIC land surface hydrological model re-calibrated against high resolution meteorological forcing data and the naturalized streamflow data along the main course of the Yellow River. As compared with the VIC model in the global hydrological forecasting system, the Nash-Sutcliffe efficiency (NSE) calculated against observed streamflow at the outlet of the Yellow River basin increases from less than 0.5 (Yuan et al., 2015) to 0.63 (Yuan, 2016). Moreover, the calibration has been done sub-basin by

sub-basin, constrained by the naturalized streamflow data from 12 hydrological gauges from upper to lower reaches. As the first companion paper, it also explores the natural hydrological predictability by using the reverse ESP simulations. Some of the key findings, as illustrated in the abstract, are as follows. (1) Difference in predictability from upper to lower reaches: for the streamflow forecasts initialized at the end of the rainy season, the influence of ICs for lower reaches of the Yellow River can be 5 months longer than that for the upper reaches, while such difference drops to 1 month during the rainy season. (2) The role of surface water ICs: the initial surface water state is the main source of streamflow predictability during the first month, beyond which other sources of terrestrial memory become more important. (3) Predictability during extremes: the dominance of ICs on the streamflow predictability can be extended by a month during the dry/wet periods, suggesting the usefulness of the ESP forecasting approach after the onset of the hydrological extreme events.

My other concern is on the revESP approach as a way to estimate the impact of IC uncertainties on hydrological forecasting. Although this approach has been used in several published studies, it is still necessary to point out that this approach significantly overestimates the uncertainty associated with IC as it uses all historical ICs. This is more so than the ESP approach for meteorological forcing. Please note that the meteorological forcing is during the forecast period which is unknown at the time of forecast, but the IC is just not able to be completely observed. The IC is the result of past meteorological conditions that have been observed to a large degree. So cautions need to be raised when interpretation of the results (ESP vs revESP), and some discussion is necessary on this issue in the end.

Response: We thank the reviewer for the comments. The revESP provides a "theoretical" framework to compare the importance of ICs and meteorological forcings in terms of hydrological predictability. In this study, both the ESP and revESP uses all historical meteorological forcings and ICs respectively, with 28 ensembles for both. However, in a real forecast, the forecaster do not necessarily use all ensemble members, i.e., both the

uncertainty (sample) in ICs and meteorological forcings can be reduced through prior information. The ESP/revESP simulation comparisons just show the major sources of predictability for a given basin, and they will guide the investment in the refinement of ICs or the improvement in climate predictions. We will add discussion in the end of the revised manuscript as follows: "2) the revESP method only assesses the theoretical predictability control by using all historical ICs. Actually, operational forecaster can refine the ICs to some extent before issuing the forecasts because of the tendency in the ICs (i.e., prior information). In this regard, the revESP may overestimate the uncertainty in the ICs. On the other hand, the ESP method may also overestimate the uncertainty in the meteorological forcings because a conditional ESP method that is based on certain teleconnections (van Dijk et al., 2013) can be used to select the meteorological forcings from all historical samples. A more elastic method that is recently proposed by Wood et al. (2016) could be used to understand the role of ICs in the seasonal hydrological forecasting with various level of uncertainty;"

Minor issues

1. Page 2 line 9: what is a more extreme climate?

Response: We will remove it to avoid confusion, and the sentence will be revised as "The intensification of the water cycle leads to an increase of hydrological extreme events..."

2. Page 2 line 11: Some references are needed to back up this statement. 3. Page 2 line 12: Some references are needed here, too.

Response: References (Huntington, 2006; Oki and Kanae, 2006) and (IPCC, 2014) will be included.

4. Page 2 line 14: There is a different between mitigation and adaption. Should seasonal forecast be more helpful with mitigation instead of adaptation? Adaptation usually happens at much longer time scales.

Response: According to the definition of IPCC. Mitigation refers to "An anthropogenic intervention to reduce the sources or enhances the sinks of greenhouse gases", and the adaption refers to "adjustment in natural or human systems to a new or changing environment". Due the inertia of the ocean that has already assimilate much carbon, the effect of reducing $CO_2$ (mitigation) on slowing the temperature will not be significant in a short time; while the adaption is an action to the changing environment or the extremes (e.g., drought and floods). And a well-planned adaptation cannot be achieved without a reliable prediction of the future.

5. Page 2 line 19: Why Atlantic Ocean? What about other oceans?

Response: We will remove it to avoid confusion, and revise the sentence as "While the decadal hydrological prediction is still at an exploring stage due to very limited predictability over land. . ."

6. Page 4, line 10: regridding usually means changing the spatial resolution of a grid data product. Here the station data is interpolated somehow to a fixed grid, so it is not regridding. It is also necessary to mention how the interpolation is done.

Response: Thanks for the comment. We will revise it as "The meteorological forcing datasets from 324 meteorological stations are interpolated into 1321 grids at a 0.25-degree resolution, with a lapse rate correction for temperature at different elevations. The observations from three nearest meteorological stations are interpolated to each grid by using the inverse quadratic distance weighting method."

7. Page 4 line 20: "river is suspended ??" What do you mean by that? I guess what you want to say is that "the riverbed is elevated above the adjacent floodplains due to sediment deposition and man-made levees".

Response: Thanks for the comment. We will revise it as ". . .where the riverbed is elevated above the adjacent floodplains due to sediment deposition and man-made levees."

8. Page 5 line 20-21: Do you have a source for these statistics?

Response: They are reported by the Bulletin of Water Resources. We will mention it in the revised manuscript.

9. Page 7 line 17: "dominant role of IC's for streamflow predictability". See the major concern #2. This interpretation needs to be cautious.

Response: Please see our response to the major concern #2 above. As suggested by the reviewer, we will replace all "dominant" with "prevails over" or "significantly contribute to" throughout the revised manuscript.

10. Page 8, line 7: what is a full initialization?

Response: It means "both the initializations of surface and subsurface water". We will mention it in the revised manuscript.

11. Figure 2: this is useful to show the spatial variation of mean temperature, precipitation and wind. But it is not the most useful ones, for example the wind is never discussed in the study. It is actually necessary to show the seasonal cycle of precipitation (and probably temperature) over the basin, just because you use such information in Figure 8.

Response: Thanks for the comment. We will revise Figure 2 to show the seasonal mean precipitation and temperature, and remove the climatology plots.

12. Figure 8: why are there a number of small streams showing the max lead time of 6 months all the time?

Response: They are caused by slow velocity. We will exclude them to focus on the results along the main courses.

References:

Huntington, T. G.: Evidence for intensification of the global water cycle: review and

synthesis, J. Hydrol., 319, 83-95, doi: 10.1016/j.jhydrol.2005.07.003, 2006.

IPCC: Summary for policymakers. In: Climate Change 2014: Impacts, Adaptation, and Vulnerability. Part A: Global and Sectoral Aspects. Contribution of Working Group II to the Fifth Assessment Report of the Intergovernmental Panel on Climate Change [Field, C.B., V.R. Barros, D.J. Dokken, K.J. Mach, M.D. Mastrandrea, T.E. Bilir, M. Chatterjee, K.L. Ebi, Y.O. Estrada, R.C. Genova, B. Girma, E.S. Kissel, A.N. Levy, S. MacCracken, P.R. Mastrandrea, and L.L. White (eds.)]. Cambridge University Press, Cambridge, United Kingdom and New York, NY, USA, pp. 1-32, 2014.

Oki, T., and Kanae, S.: Global hydrological cycles and world water resources, Science, 313, 1068-1072, 2006.

Wood, A. W., Hopson, T., and Newman, A., et al.: Quantifying streamflow forecast skill elasticity to initial condition and climate prediction skill, J. Hydrometeorol., 17, 651-668, doi: 10.1175/JHM-D-14-0213.1, 2016.

Yuan, X., Roundy, J. K., Wood, E. F., and Sheffield, J.: Seasonal forecasting of global hydrologic extremes: system development and evaluation over GEWEX basins, Bull. Am. Meteorol. Soc., 96, 1895-1912, doi:10.1175/BAMS-D-14-00003.1, 2015.

Yuan, X.: An experimental seasonal hydrological forecasting system over the Yellow River basin – Part II: The added value from climate forecast models, Hydrol. Earth Syst. Sci. Discuss., doi:10.5194/hess-2016-102, in review, 2016.

---

## Author Response (AR1)

[revised manuscript text omitted]

Email: yuanxing@tea.ac.cn
Tel: +86-10-82995385
http://www.escience.cn/people/yuanxing
May 22, 2016

Dr. Alexander Gelfan
Editor
Hydrology and Earth System Sciences

15   RE: manuscript #hess-2016-101

Dear Dr. Gelfan,

Thank you for your kind decision letter on our manuscript entitled "An experimental seasonal
20   hydrological forecasting system over the Yellow River basin-Part I: Understanding the role of initial hydrological conditions" (hess-2016-101). We have carefully considered your and reviewer's comments and incorporated them into the revised manuscript to the extent possible. We hope that you find the revised manuscript and the response to the reviews acceptable to *HESS*.
The detailed responses to the comments are attached.

We appreciate the effort you spent to process the manuscript and look forward to hearing from you soon.

Sincerely yours,

30   Xing Yuan

**Responses to comments from Dr. V. Moreydo**

*A set of two papers by Xing Yuan is presented for the review. The first one is devoted to the investigation of the role of initial watershed conditions for streamflow forecasting in the Yellow River basin, China. The second paper deals with seasonal ensemble forecasting of the human-induced streamflow by the means of North American Multimodel Ensemble (NMME) and forecast post-processing procedure. Given a vast material and several main issues under consideration within a single modeling framework, splitting up the papers is a good decision. This comment addresses the first paper of the two. The paper investigates the predictability of streamflow and soil moisture on a seasonal extent (up to 6 months ahead) within a large river basin, characterized with different streamflow formation patterns. Seasonal streamflow predictability studies are considered to be one of the major directions for the hydrological scientific community (Blöschl, 2006), and the presented paper is aimed to outline the regional patterns of predictability, thus fully corresponding with the international hydrological agenda. The authors assess contributions of initial watershed conditions and meteorological forcings within the forecast window to seasonal predictability of the streamflow along the main course of the Yellow River. To fulfill the task, an experiment is designed based on the Reverse-Ensemble streamflow prediction method (rev-ESP, Wood and Lettenmaier, 2008). The Variable infiltration capacity (VIC) hydrological model (Liang et al., 1996) with global routing model by Yang et al. (2015) is used to describe the initial conditions and simulate the streamflow for the test period of 1982 – 2010. Meteorological forcings are set on a grid-base with 0.25 degree resolution holding the interpolated data from 324 meteorological stations within the basin. The hydrological model is calibrated against naturalized streamflow (reconstructed for the whole basin with the rainfall data using rainfall-runoff ratio at 12 main river gauges) and shows overall good performance with NSE values of 0.78 – 0.91. One of the issues the reviewer would like to address concerning the experiment is that the naturalized streamflow may contain errors concerning the precipitation and streamflow measurements. Reaching up to 10% of the measured value, the errors, when combined, might introduce uncertainties in the naturalized streamflow time-series. The uncertainties may be transferred forward to the hydrological model by calibrating it against these time-series which may result in unrealistic parameters. It should be noted that the authors use naturalized streamflow in order to assess natural and unperturbed hydrological predictability apart of the human induced water subtractions. The observed streamflow is used in the second paper to condition the hydrological forecasts. The overall design of the experiment complies with the ESP/revESP framework. For the ESP experiments the model is first run from 1951 to 2010 to simulate the natural initial conditions of the basin, then for the ESP experiment the simulation is initialized at the beginning of each calendar month within the period 1982 – 2010 with 28 6-month-long realizations of meteorological forcing. For the revESP experiment the model is run again for 6 months starting each month within the test period with single actual meteorological forcing, but the initial conditions are taken from the ensemble of 28 members excluding the year under consideration. The memory of the hydrological system is demonstrated by calculating the ratio of the RMSE from ESP and revESP experiments, respectively. The ratio values less than 1 show initial conditions prevailing over meteorological forcing in streamflow predictability over the forecast period and the memory of the hydrological system is controlling the streamflow conditions. The ratio greater than 1 shows that the predictability depends on climatic system and the memory of the*

*hydrological system has no control over the streamflow. Results obtained for 12 gauges along the river course show that the initial conditions control the system in Autumn and Winter, while meteorological conditions tend to influence the system in the Spring and Summer rainy season. The authors outline that it has been discovered that the hydrological system memory drops to the lowest level at the end of the*
5 *dry season. Further investigation of the initial surface water conditions is also carried out, concluding that the system memory of the surface water state lasts for less than a month in general and has no influence on the long lead-time, yet for the short leads it is crucial. Next, the initial conditions are selected to attribute for a wet and dry state of the basin, thus tending the model to reproduce extremes. This part concludes that the wet state of the basin is generally more influenced by the initial conditions,*
10 *while the drier case is less stable. To assess the predictability of the soil moisture the same ESP/revESP technique is applied, showing that again the best forecast performance for such a slow process as soil moisture transport is obtained for the dry winter season and a significant lead-time up to 2 – 3 months can be still obtained for summer months. The overall characteristic of the paper is very good, it is a major contribution to global investigations of the limits of hydrological predictability. The paper can be*
15 *published without any revisions.*

**Response**: We would like to thank Dr. Moreydo for the compliment and recognizing the value of our work. The thoughtful comments have helped improve the manuscript. The reviewer's comments are italicized and our responses immediately follow.

20 *One of the issues the reviewer would like to address concerning the experiment is that the naturalized streamflow may contain errors concerning the precipitation and streamflow measurements. Reaching up to 10% of the measured value, the errors, when combined, might introduce uncertainties in the naturalized streamflow time-series. The uncertainties may be transferred forward to the hydrological model by calibrating it against these time-series which may result in unrealistic parameters.*

**Response**: Thanks for this important comment. We have incorporated a discussion about the uncertainty into the revised manuscript as follows:

"It should be noted that the naturalized streamflow may contain errors from the measurement of precipitation and/or streamflow, and the errors may result in uncertainty in the calibrated parameters
30 and the hydrological model. In the future, multisource (e.g., satellite and ground) observations combined with data assimilation techniques are needed to quantify such uncertainty." (P6, L8-11 in the tracked version of the revised manuscript)

**Responses to comments from Referee #2**

We are very grateful to the reviewer for the positive and careful review. The thoughtful comments have helped improve the manuscript. The reviewer's comments are italicized and our responses immediately
5  follow.

*This manuscript presents a seasonal hydrological forecast system for the Yellow River basin and*
*investigates the contribution of hydrological initial condition and meteorological forcing to the*
*predictability of soil moisture and streamflow over the study region. The topic is suitable for HESS, and*
10  *the research method is scientifically sound. The manuscript is generally well written with good quality*
*illustrations. It is a good piece of work. But I am a little disappointed with the scientific contribution of*
*this work to our knowledge and understanding about seasonal hydrological forecasting, at least in the*
*way that was presented. This is one of my major concerns. There are also a number of places in the*
*manuscript that needs clarification or justification. Overall, I think a major revision is necessary to*
15  *improve the quality of the paper.*

**Response**: We thank the reviewer for the comments. The comments have helped to improve the manuscript a lot. Please see our clarification of the novelty and the responses to the comments below.

20  *Major concerns:*

*This is a solid piece of research work, but there isn't really anything new in terms of research methods*
*or scientific understanding. Several previous studies have adopted the exact same methodology and*
*answered the exact same questions, but just over different basins. So besides applying the same*
25  *methodology in a forecast system over the Yellow River basin (new to some degree), there is not enough*
*evidence to support the novelty of this research. The authors argue that a new meteorological dataset*
*with higher resolution is used, but it was not demonstrated how this improved resolution actually help*
*with the hydrological forecasting.*

30  **Response**: We thank the reviewer for the comments. The two companion papers introduce an experimental seasonal hydrological forecasting system over the Yellow River basin in northern China to provide adaptive support in a changing environment. The system draws from a legacy of a global hydrological forecasting system (Yuan et al., 2015), but with the VIC land surface hydrological model re-calibrated against high resolution meteorological forcing data and the naturalized streamflow data
35  along the main course of the Yellow River. As compared with the VIC model in the global hydrological forecasting system, the Nash-Sutcliffe efficiency (NSE) calculated against observed streamflow at the outlet of the Yellow River basin increases from less than 0.5 (Yuan et al., 2015) to 0.63 (Yuan, 2016).

Moreover, the calibration has been done sub-basin by sub-basin, constrained by the naturalized streamflow data from 12 hydrological gauges from upper to lower reaches.

As the first companion paper, it also explores the natural hydrological predictability by using the reverse ESP simulations. Some of the key findings, as illustrated in the abstract, are as follows.

(1) **Difference in predictability from upper to lower reaches**: for the streamflow forecasts initialized at the end of the rainy season, the influence of ICs for lower reaches of the Yellow River can be 5 months longer than that for the upper reaches, while such difference drops to 1 month during the rainy season. (P1, L27-28 in the tracked version of the revised manuscript)

(2) **The role of surface water ICs**: the initial surface water state is the main source of streamflow predictability during the first month, beyond which other sources of terrestrial memory become more important. (P1, L30-31 in the revised manuscript)

(3) **Predictability during extremes**: the dominance of ICs on the streamflow predictability can be extended by a month during the dry/wet periods, suggesting the usefulness of the ESP forecasting approach after the onset of the hydrological extreme events. (P1, L31-33 in the revised manuscript)

*My other concern is on the revESP approach as a way to estimate the impact of IC uncertainties on hydrological forecasting. Although this approach has been used in several published studies, it is still necessary to point out that this approach significantly overestimates the uncertainty associated with IC as it uses all historical ICs. This is more so than the ESP approach for meteorological forcing. Please note that the meteorological forcing is during the forecast period which is unknown at the time of forecast, but the IC is just not able to be completely observed. The IC is the result of past meteorological conditions that have been observed to a large degree. So cautions need to be raised when interpretation of the results (ESP vs revESP), and some discussion is necessary on this issue in the end.*

**Response**: We thank the reviewer for the comments. The revESP provides a "theoretical" framework to compare the importance of ICs and meteorological forcings in terms of hydrological predictability. In this study, **both the ESP and revESP uses all historical meteorological forcings and ICs respectively, with 28 ensembles for both**. However, in a real forecast, the forecaster do not necessarily use all ensemble members, i.e., both the uncertainty (sample) in ICs and meteorological forcings can be reduced through prior information. The ESP/revESP simulation comparisons just show the major sources of predictability for a given basin, and they will guide the investment in the refinement of ICs or the improvement in climate predictions. We have added some discussions in the end of the revised manuscript as follows:

"2) the revESP method only assesses the theoretical predictability control by using all historical ICs. Actually, operational forecaster can refine the ICs to some extent before issuing the forecasts because of the tendency in the ICs (i.e., prior information). In this regard, the revESP may overestimate the uncertainty in the ICs. On the other hand, the ESP method may also overestimate the uncertainty in the

meteorological forcings because a conditional ESP method that is based on certain teleconnections (van Dijk et al., 2013) can be used to select the meteorological forcings from all historical samples. A more elastic method that is recently proposed by Wood et al. (2016) could be used to understand the role of ICs in the seasonal hydrological forecasting with various level of uncertainty;" (P12, L6-12 in the revised manuscript)

*Minor issues*

*1. Page 2 line 9: what is a more extreme climate?*

**Response**: We have removed it to avoid confusion, and the sentence has been revised as "The intensification of the water cycle leads to an increase of hydrological extreme events…" (P2, L9 in the revised manuscript)

*2. Page 2 line 11: Some references are needed to back up this statement.*

*3. Page 2 line 12: Some references are needed here, too.*

**Response**: References (Huntington, 2006; Oki and Kanae, 2006) and (IPCC, 2014) have been included. (P2, L11 & L14 in the revised manuscript)

*4. Page 2 line 14: There is a different between mitigation and adaption. Should seasonal forecast be more helpful with mitigation instead of adaptation? Adaptation usually happens at much longer time scales.*

**Response**: According to the definition of IPCC. Mitigation refers to "An anthropogenic intervention to reduce the sources or enhances the sinks of greenhouse gases", and the adaption refers to "adjustment in natural or human systems to a new or changing environment". Due to the inertia of the ocean that has already assimilated much carbon, the effect of reducing $CO_2$ (mitigation) on slowing the temperature will not be significant in a short time; while the adaption is an action to the changing environment or the extremes (e.g., drought and floods) with a shorter time scale than the mitigation. And a well-planned adaptation cannot be achieved without a reliable prediction of the future.

*5. Page 2 line 19: Why Atlantic Ocean? What about other oceans?*

**Response**: We have removed it to avoid confusion, and revised the sentence as "While the decadal hydrological prediction is still at an exploring stage due to very limited predictability over land…" (P2, L19-20 in the revised manuscript)

5  6. *Page 4, line 10: regridding usually means changing the spatial resolution of a grid data product. Here the station data is interpolated somehow to a fixed grid, so it is not regridding. It is also necessary to mention how the interpolation is done.*

**Response**: Thanks for the comment. We have revised it as "The meteorological forcing datasets from
10  324 meteorological stations are interpolated into 1321 grids at a 0.25-degree resolution, with a lapse rate correction for temperature at different elevations. The observations from three nearest meteorological stations are interpolated to each grid by using the inverse quadratic distance weighting method." (P4, L10-12 in the revised manuscript)

15  7. *Page 4 line 20: "river is suspended ??" What do you mean by that? I guess what you want to say is that "the riverbed is elevated above the adjacent floodplains due to sediment deposition and man-made levees".*

**Response**: Thanks for the comment. We have revised it as "…where the riverbed is elevated above the
20  adjacent floodplains due to sediment deposition and man-made levees." (P4, L22-23 in the revised manuscript)

8. *Page 5 line 20-21: Do you have a source for these statistics?*

25  **Response**: They were reported by the Bulletin of Water Resources, but there were some typos. We have corrected it in the revised manuscript as follows:

"As reported by the Bulletin of Water Resources, the observed annual mean streamflow at the outlet of the basin (i.e., Lijin station) is about $3.15 \times 10^{10}$ m$^3$ during 1956-2000, while the annual mean consumed and inter-basin diverted water is $1.48 \times 10^{10}$ m$^3$." (P5, L22-24)

9. *Page 7 line 17: "dominant role of IC's for streamflow predictability". See the major concern #2. This interpretation needs to be cautious.*

**Response**: Please see our response to the major concern #2 above. As suggested by the reviewer, we have replaced all "dominant" with "prevails over" or "significantly contribute to" throughout the revised manuscript.

5 *10. Page 8, line 7: what is a full initialization?*

**Response**: It means "both the initializations of surface and subsurface water". We have mentioned it in the revised manuscript. (P8, L16)

10 *11. Figure 2: this is useful to show the spatial variation of mean temperature, precipitation and wind. But it is not the most useful ones, for example the wind is never discussed in the study. It is actually necessary to show the seasonal cycle of precipitation (and probably temperature) over the basin, just because you use such information in Figure 8.*

15 **Response**: Thanks for the comment. We have revised Figure 2 to show the seasonal mean precipitation and temperature, and removed the climatology plots. (Figure 2 in the revised manuscript)

*12. Figure 8: why are there a number of small streams showing the max lead time of 6 months all the time?*

**Response**: They are caused by slow velocity. We have excluded them to focus on the results along the main courses. (Figure 8 in the revised manuscript)

---

## Editor Decision (ED1)

The authors have taken into account the criticisms and suggestions of both referees that resulted in improvement of the paper. I recommend the revised paper for publication as is.

| Principal Criteria | Excellent (1) | Good (2) | Fair (3) | Poor (4) |
|---|---|---|---|---|
| **Scientific Significance:** Does the manuscript represent a substantial contribution to scientific progress within the scope of *Hydrology and Earth System Sciences* (substantial new concepts, ideas, methods, or data)? | | + | | |
| **Scientific Quality:** Are the scientific approach and applied methods valid? Are the results discussed in an appropriate and balanced way (consideration of related work, including appropriate references)? | | + | | |
| **Presentation Quality:** Are the scientific results and conclusions presented in a clear, concise, and well-structured way (number and quality of figures/tables, appropriate use of English language)? | | + | | |

Alexander Gelfan

Handling Editor